# New Activated Carbon from Mine Coal for Adsorption of Dye in Simulated Water or Multiple Heavy Metals in Real Wastewater

**DOI:** 10.3390/ma13112498

**Published:** 2020-05-30

**Authors:** Marwa Elkady, Hassan Shokry, Hesham Hamad

**Affiliations:** 1Fabrication Technology Research Department, Advanced Technology and New Materials Research Institute (ATNMRI), City of Scientific Research and Technological Applications (SRTA-City), New Borg El-Arab City 21934, Egypt; 2Chemical and Petrochemical Engineering Department, Egypt-Japan University of Science and Technology (E-JUST), New Borg El-Arab City 21934, Egypt; 3Electronic Materials Researches Department, Advanced Technology and New Materials Research Institute (ATNMRI), City of Scientific Research and Technological Applications (SRTA-City), New Borg El-Arab City 21934, Egypt; hassan.shokry@gmail.com; 4Environmental Engineering Department, Egypt-Japan University of Science and Technology (E-JUST), New Borg El-Arab City 21934, Egypt

**Keywords:** batch adsorption, nano-activated carbon, El-Maghara mine coal, petrochemical waste water, mechanism

## Abstract

Nano-activated carbon (NAC) prepared from El-Maghara mine coal were modified with nitric acid solution. Their physico-chemical properties were investigated in terms of methylene blue (MB) adsorption, FTIR, and metal adsorption. Upon oxidation of the AC_S_ with nitric acid, surface oxide groups were observed in the FTIR spectra by absorption peaks at 1750–1250 cm^−1^. The optimum processes parameters include HNO_3_/AC ratio (wt./wt.) of 20, oxidation time of 2 h, and the concentration of HNO_3_ of 10% reaching the maximum adsorption capacity of MB dye. Also, the prepared NAC was characterized by SEM, EDX, TEM, Raman Spectroscopy, and BET analyses. The batch adsorption of MB dye from solution was used for monitoring the behavior of the most proper produced NAC. Equilibrium isotherms of MB dye adsorption on NAC materials were acquired and the results discussed in relation to their surface chemistry. Langmuir model recorded the best interpretation of the dye adsorption data. Also, NAC was evaluated for simultaneous adsorption of six different metal ions (Fe^2+^, Ni^2+^, Mn^2+^, Pb^2+^, Cu^2+^, and Zn^2+^) that represented contaminates in petrochemical industrial wastewater. The results indicated that the extracted NAC from El-Maghara mine coal is considered as an efficient low-cost adsorbent material for remediation in both basic dyes and metal ions from the polluted solutions.

## 1. Introduction

The availability of clean water is getting scarce due to world industrialization. Nowadays, the world is suffering from a shortage of clean water, particularly in the developing countries. Water pollutants may be either organics such as dyes, phenolic compounds, and heavy metal ions such as lead, arsenic, and zinc. All of these pollutants are characterized by their non-biodegradable nature posing a great risk to human health and the global environment [1]. The textile manufacture process consumes large water quantities and represents one of the major sources for discharging huge amounts of wastewater that increases environmental danger. This industry uses more than ten thousand commercial products, among them, more than three thousand are dyes [2,3]. As well, the petrochemical industry poses issues related to a lot of wastewater that is contaminated with different heavy metals. Owing to the great degree of the stability and non-biodegradable nature of these wastewater contaminates, dyes and heavy metals create a serious problem to the environment [4]. Therefore, the remediation of dyes and heavy metal ions from wastewater represents an urgent need for environmental concerns [5].

Investigation into the specialized processes has been developed for the removal of metal ions and/or dyes from waste discharges. The technological systems include chemical precipitation, adsorption, photo-catalysis, ion exchange, cementation, ozone oxidation, membrane separation, polymer-enhanced ultrafiltration, and biological and electrochemical operations [6,7,8,9,10,11,12,13,14,15,16,17]. Among various treatment technologies, adsorption techniques have proven to be effective, reliable, easy to operate, insensitive to toxic materials, and economic [18,19].

Recently, utilization of low-cost adsorbent materials for water remediation was researched by numerous scientists. Activated carbon (AC) is a highly effective adsorbent for eliminating heavy metals and dyes from polluted solutions due to its appropriate physical and chemical characteristics. AC is characterized by its high specific surface area, several surface functional groups, well-developed internal pore structure, low density, good mechanical strength, easy regeneration, good chemical and thermal stability, and suitability for large scale production, which can be modified by the chemical method in presence of a specific solvent [15,20,21]. AC has recently attracted great attention in water and wastewater applications due to its cost-effectiveness, abundance, renewability, suitable absorptive properties, and environmental friendliness [22,23]. 

There are many sources of AC from many carbonaceous precursors including tires [24], de-oiled soya [25], rubber, fly ash [26], sugar industrial wastes [26], orange peel [27], coal [15], and scrap tire [28]. Among these resources, the highest promising activated carbon extracted is from coal. It is classified into four general categories and its percentage of carbon and heat values (H.V) with the unit of British Thermal Unit (BTU) are shown in Table 1 [29,30]. The most popular carbons are bituminous based materials. These materials characterized by their abrasion resistance, hardness, good porosity and cost cheapness, but their effectiveness needs to be tested in each application to optimize the product.

Usually, AC has an aromatic molecular structure of graphene layers and oxygen-containing functional groups [29]. Several techniques were tested to enhance their structure by introducing other hetero-atom-containing functional groups. The selectivity and adsorptive capacity of the classical ACs towards the pollutant can be effectively improved by the introduction of weakly functional groups that use oxidizers such as nitric acid, sulfuric acid, permanganate, bichromate, and hydrogen peroxide [30]. The proportion of the weak surface functional groups of the acid is an important factor in the selective removal of contaminants from the polluted solutions [31].

The aim of the activated carbon surface oxidation process is obtaining a more hydrophilic and polar surface structure that includes a lot of oxygen-containing surface groups. The surface functional groups anchored within carbons are responsible for a variety of physico-chemical and catalytic activities of materials [32]. Enhancing the surface chemistry of porous carbons represents a challenge toward discovering the novel applications of these materials [33]. There is a difference in metal ion decontamination on the various carbon materials that depend upon the forms of the metal complexes and the oxygen-containing surface groups on AC. 

The claim of the present investigation is to develop a new adsorbent with low cost and high efficiency via oxidation by nitric acid treatment to produce oxygen-containing functional groups and increase the hydrophilicity of the surface carbon and following their adsorption performance for both heavy metal ions from industrial wastewater of petrochemicals and methylene blue adsorption from aqueous solutions. The adsorption effectiveness of the resulting carbon material will be evaluated toward the different cations to elucidate its adsorption selectivity against both heavy metals and dye molecules. 

## 2. Materials and Methods 

### 2.1. Materials

Maghara coal that was obtained from El-Maghara field coal, located in North-Sinai Governorate, Egypt, was used for preparing nano-activated carbons (NACs). The chemical analysis of the coal is investigated in Table 2. Methylene blue is selected as cationic basic dye model supplied from Nice Chemicals Pvt. Ltd., Kochi, Kerala, India. Methylene blue (MB) dye is the most generally used substance for dyeing cotton, wood, and silk [34]. The heavy metal (Fe^2+^, Zn^2+^, Pb^2+^, Mn^2+^, Cu^2+^, and Ni^2+^) polluted wastewater delivered from the Sidi-Kerir Petrochemicals Company (Alexandria Governorate, Egypt) is treated using El-Maghara coal as raw coal (RC) and optimum prepared nano-activated carbon. 

### 2.2. Preparation of Nano-Activated Carbon (NAC)

Various activated carbon samples (AC) were prepared from the El-Maghara coal by the chemical activation method through mixing the coal with 50% sodium hydroxide solution and carbonizing this mixture at 550 °C for 90 min using muffle furnace (FM-36, Yamato, Japan). Chemical treatment of activated carbon extracted from El-Maghara coal was performed using HNO_3_ solution as an oxidizing agent. The coal was pretreated with nitric acid solutions at concentrations in the range of 10% to 40% and heated at the boiling point of nitric acid (115 °C) for 1 to 3.5 h. The nitric acid-treated sample was washed by centrifugation (5000 rpm, 10 min) three times using hot water until it reached a constant pH of 6.5. Finally, the resultant powders were dried at 50 °C for 24 h. All prepared NAC samples were monitored toward MB adsorption. The optimum prepared activated carbon sample that attained the highest MB dye adsorption efficiency was characterized to determine its physico-chemical properties. The schematic of formation of the NAC from El-Maghara mine coal under the optimized conditions of chemical activation is illustrated in Figure 1a. 

### 2.3. Batch Mode Adsorption Experiment

The effectiveness of the nitric acid-treated AC materials was tested for adsorption of MB dye. The static equilibration models were used to determine adsorption isotherms of the dye adsorption process. A known quantity of 0.2 g/L from the prepared AC sample was placed in a series of 250 mL flasks containing 200 mL of dye solution at various initial concentrations in the range of 0 to 30 ppm at natural pH (6.8–7.1) without any pH adjustment. The flasks were then shaken at constant temperature (25 ± 2 °C) until the equilibrium obtained after 1 h, as evident from Table 3. Subsequently, the carbon materials were removed from the treated solution by centrifugation at 5000 rpm for 5 min. Adsorbate concentrations in the filtrate were determined using UV-Spectrophotometer (Libra S.11, Wolf Laboratories Limited, Pocklington, UK) at 665 nm as λ_max_. 

### 2.4. Determination of Heavy Metals Adsorption Capacity from Real Wastewater

The performance of the optimum prepared NAC that recorded the maximum dye adsorption capacity was compared with raw El-Maghara coal (RC) for removal of six heavy metal ions (Fe^2+^, Zn^2+^, Pb^2+^, Mn^2+^, Cu^2+^, and Ni^2+^) from petrochemicals wastewater as indicated in Figure 1. The schematic performance of the prepared NAC that was obtained from the optimum operational parameters for heavy metal decontamination from real petrochemical wastewater is shown in Figure 1a. The comparable investigation of heavy metal sorption affinity for both NAC and RC was evident from Figure 1b. The polluted wastewater was collected at 28 °C and its initial degree of acidity was measured as 7.8. The batch technique was applied through mixing 0.2 g from each carbon sample separately with 100 mL of wastewater for a selected contact time of 2 h at room temperature (25 ± 2) °C. This selected contact time interval used for determination of the adsorption behavior of NAC toward the multi-metal ions was based upon the fact that the presence of multi-metal together at the adsorption process hindered and competed with each other to be reached and adsorbed onto the material active sites. So, the contact time interval for testing the adsorption behavior of NAC toward the multi-metal ions was improved compared with that studied for dye adsorption process to be able to differentiate between the selectivity of the material towards the different studied metals. Following the adsorption step, the NAC sample and the liquid phase were separated by centrifugation. The preliminary and final concentrations of heavy metals using each tested carbon material were determined by the atomic absorption spectrometer (Perkin Elmer 2380 Atomic Absorption Spectrometer, Waltham, MA, USA). The amount of adsorbate (dye or heavy metals) per unit weight of adsorbent, *q_e_* (mg/g), was calculated by Equation (1)
(1)$$q_{e}=\frac{\left(C_{i}-C_{e}\right)V}{m}$$
where *C_i_* and *C_e_* are the initial and the equilibrium concentrations of adsorbate (mg/L), *V* is solution volume (L) and *m* is activated carbon dry weight (g).

The dye adsorption equilibrium data were simulated using the Langmuir and Freundlich isotherm models, in order to determine the interactions between the nitric acid modified AC and cation pollutants.

### 2.5. Adsorption Isotherms

Langmuir and Freundlich isotherms were extensively used to determine the adsorption profile of the studied process. These isotherms are essential for explanation of the interaction of harmful ions with the optimum prepared NAC sample. Therefore, empirical equations (Freundlich or Langmuir isotherm model) are important to interpret and predict the adsorption data. The experimental results of the dye adsorption process were modeled using both Freundlich and Langmuir equations [34].

The assumption of Langmuir model is dependant upon the formation of a monolayer from adsorbate onto the adsorbent surface at a specific number of definite sites. The linearization of Langmuir equation is given by:*C_e_*/*q_e_* = 1/(*k_a_q_m_*) + *C_e_*(1/*q_m_*)(2)

Where *q_e_* is the amount adsorbed (mg/g), Ce is the equilibrium concentration of the adsorbate ions (mg/L), and *q_m_* and *k_a_* are Langmuir constants related to maximum adsorption capacity (monolayer capacity) (mg/g) and energy of adsorption (L/mg), respectively. 

The Freundlich isotherm attempts to account for surface heterogeneity, and its linear form is given by:Log *q_e_* = log *K_f_* + (1/*n*) log *C_e_*(3)
where *K_f_* and *n* are Freundlich constants related to adsorption capacity and adsorption intensity, respectively.

### 2.6. Determination the Physico-Chemical Properties of Prepared Materials

The morphological and chemical structure of the optimum prepared NAC that attained high adsorption affinity toward methylene blue were established using Transmission Electron Microscope (TEM, JEM-2100, Tokyo, JAPAN), which was associated with Energy Dispersive X-Ray Spectroscopy (EDX) for quantitative elemental analysis. Scanning Electron Microscope (SEM) (JEOL JSM 6360LA, Tokyo, Japan) was used to confirm and support the TEM micrograph. Moreover, the structure configurations of the prepared sample NAC were recorded using Raman spectra (SENTERRA spectrometer Bruker, Karlsruhe, Germany) at different locations with a 532 nm Ar laser with spot size of 105 µm. Surface area of prepared sample NAC was compared with El-Maghara coal through samples degassed at 150 °C for 2 h under a dry N_2_ gas purge prior to N_2_ adsorption measurements at Belsorbmini II, BEl Japan Inc., Osaka, Japan. The point of zero charge (pHpzc) of prepared NAC that represents the material pH at which the net charge of total absorbent’s surface is equal to zero was determined experimentally through equilibrium 0.15 g of sample with 50 mL of 0.1 molar NaCl. The solution pH was adjusted to remain within the range between 1 and 12 using 0.01 M NaOH and/or 0.01 M HCl and the mixture was stirred for 24 h at 150 rpm at room temperature. After completing the mixing time, the pH of the mixture solution was measured using digital pH meter model 3305 of (Jenway Ltd., Dunmow, Essex, UK).

Finally, the main functional groups onto both El-Maghara coal and the prepared NAC material surfaces were examined using FTIR-8400S Shimadzu, Kyoto, Japan. Also, the material interaction with dye molecules after dye adsorption process was identified using FTIR. All spectra were recorded using a KBr pellet containing 0.1% of the carbon samples. Before each measurement, the instrument was run to collect the background, which was automatically subtracted thereafter from the sample spectrum. IR transmission spectra were recorded in the range from 400 to 4000 cm^−1^ with scan rate 2 cm^−1^ s^−1^.

## 3. Results and Discussion

### 3.1. Structural Identification and Characterization of Activated Carbon Derived from El-Maghara Mine Coal

The surface functional groups (chemical structure) of AC materials have a significant impact on their adsorption characteristics, so, the FT-IR spectra for El-Maghara mine coal and the prepared NAC were compared at Figure 2a,b, respectively. The peaks presented at the two carbon samples around 2921, 2851, and 1376 cm^−1^ can be assigned to various C–H bonds [35,36]. The very weak bands at the two carbon samples near 2300 cm^−1^ are assigned to carbon–oxygen groups due to a ketone [37]. On comparing the spectra for the two carbon samples, a new specific and strong peak emerged at 1640 cm^−1^ observed at prepared NAC. This peak is assigned to the plentiful carbonyl C=O stretching groups by carboxylic acid or lactone groups that represents nitric acid oxidation [38]. The high intensity of this peak indicated that the oxidation treatment produces more acidic sites and the strong character of nitric acid reduces the carbon surface electrons inducing partial positive charges. In addition, the surface oxides were formed through adsorption the solution oxygen anions [39] that reduces the Lewis basicity of the carbon surface. This result indicated the enriched surface of the activated carbons prepared from oxidation with nitric acid could significantly enhance their adsorption performance. The peak at 1557 cm^−1^ gives a prediction about the inclusion of natural coal with some amide groups that disappeared after nitric acid modification at the prepared activated carbon [40]. The new peak appeared at 1060 cm^−1^ after nitric acid modification was assigned to the presence of asymmetric NO_2_ stretch vibration that confirm carbon surface modification [41]. 

The peaks presented at the two carbon materials at 685, 1060, and 1250 cm^−1^ are assigned to the aromatic ring’s presence [42]. The broad peaks around 3400–3800 cm^−1^ and 1557–1640 cm^−1^ at the two carbon samples spectra were ascribed at O–H stretching vibration involved in hydrogen bonds due to the existence of chemisorbed water and surface hydroxyl groups that are responsible for dye adsorption interaction [42]. 

The morphological and configuration structure beside chemical composition of the optimum prepared NAC were examined using TEM, SEM, Raman, and EDX techniques. A TEM micrograph is illustrated at Figure 3a, it was clearly shown that the prepared NAC has irregular agglomerates of nanoparticles with spherical shape. An SEM micrograph in Figure 3b confirms the TEM result of the NAC nanostructure. The average diameter of the nanoparticles was estimated as 20 nm, as evident from particle size distribution (Figure 3a). EDX analysis presented at Table 4 indicated that the chemical composition of the produced NAC is mainly carbon.

On the other hand, the specific structural nature of the produced NAC can be further elucidated by the Raman spectroscopy. In Figure 3c, the peaks present around 1320 cm^−1^ (D-bands) are due to the breathing mode of k-point phonons with A_1g_ symmetry, corresponding to the disordered carbon or defective graphitic structures [43]. 

The peaks around 1592 cm^−1^ (G-bands) are assigned to the E_2g_ phonon of sp^2^ carbon atoms [44], which are distinguishing graphitic layers and returned to the tangential vibration of the carbon atoms. In order to investigate the enhancement at the surface before and after modification by nitric acid, the BET measurements of El-Maghara coal and the prepared NAC were compared. Also, the porosities of the raw coal and prepared NAC were examined by the N_2_ sorption measurements Brunauer–Emmett–Teller (BET) specific surface areas. It was indicated that prepared NAC has greater specific surface area of 932 m^2^/g compared with 260 m^2^/g for raw El-Maghara coal. Moreover, the average pore volume of prepared NAC, which was determined as 0.51 cm^3^/g, is smaller than that of raw coal that measured as 1.08 cm^3^/g. This result implies that nitric acid modification process increases the material surface area of produced NAC and its porosity improves the efficiency of the adsorption process.

### 3.2. Effect of Oxidation Treatment Conditions on MB Adsorption 

Figure 4a shows the effect of nitric acid/AC on the adsorption uptake of MB. It may be predicted that there is no significant influence of the adsorption capacity of the samples with increasing nitric acid/AC ratio (wt./wt.). Almost no difference at material adsorption capacity was detected at the various studied ratios. Accordingly, the optimum selected HNO_3_/AC impregnation ratio is proposed at 20 for effective activation of El-Maghara coal with minimum consumption of activating agent. Moreover, it was stated from the literature that excess of HNO_3_ motivates a powerful gasification reaction that leads to the destruction of the carbon framework and decreases dramatically the accessible area. In addition, the excess of HNO_3_ molecules may be deposited in the carbon pore wall that may be stopping the catalytic oxidation, decreasing the adsorption uptake [45]. 

The adsorption isotherms of MB solution onto prepared NAC adsorbent at various oxidation reaction times are shown in Figure 4b. It was indicated that for the applied MB concentrations that NAC adsorption capacity was improved with increasing the oxidation time from 1 to 2 h. As the oxidation time increment went above 2 h, the material adsorption capacity tended to be decreased with increasing the oxidation time from 2 to 3.5 h. Thus, the optimum oxidation time is considered as 2 h using 30% nitric acid concentration. 

Isotherms of MB dye solution onto the prepared NAC at various nitric acid concentrations are shown in Figure 4c. It is shown that, the materials adsorption capacity was reduced as the nitric acid concentration increased. In general, the adsorption interaction between the activated carbon material and adsorbed molecules may takes place through Van der Waals interaction, hydrogen bonding, and/or electrostatic interaction according to the type of functional groups at the activated carbon and type of adsorbed molecules [46].

The number of carboxylic groups of functionalized NAC material significantly increased with the increase of the concentration of nitric acid. This result was confirmed previously by many researchers that verified oxygen containing surface functional groups amount is highly reliant on the reagents concentration [47]. While the oxygen content of AC usually increases with increasing reagent concentration, the surface area and pore volume values were adversely decreased. This may be due to the introduction of a significant number of oxygenated acidic surface complexes onto the carbon surface, these complexes postulated partial or complete pore blockage as will be confirmed at the next characterization section [48]. 

Therefore, the conditions of the chemical modification process should be identified according to the surface chemistry and textural characteristics of the material due to most of AC applications being highly connected with these properties. 

### 3.3. Adsorption Equilibrium

The previous results indicated that the optimum conditions for the nitric acid modification process were recorded at 10% nitric acid concentration with 20 HNO_3_/AC mixing ratio and 2 h oxidation time. Both Freundlich and Langmuir equations were employed to determine the maximum dye adsorption capacities of the optimum prepared NAC. The data represented in Figure 5a indicates that the dye adsorption fitted well to the Langmuir isotherm model, since R^2^ is more than 0.9 (0.96). The slope (1/*q_m_*) which is known to be the adsorption intensity, the intercept (1/*k_a_q_m_*) that determine the monolayer adsorption capacity and the correlation coefficient were determined from Figure 3 and the data is tabulated in Table 4. The maximum mono-linear adsorption capacity of MB dye onto the prepared NAC was calculated as 29.50 mg/g, which is homogeneous to the previously determined maximum theoretical adsorption capacity of MB. So, Langmuir isotherm provides the best fit for MB adsorption onto NAC. The Langmuir adsorption model deviates significantly in many cases (especially for multi-metal adsorption), primarily because it fails to account for the surface roughness of the adsorbent. Rough and inhomogeneous surfaces have multiplied site-types available for adsorption, and some parameters vary from site to site, such as the heat of adsorption. 

The Freundlich isotherm plotted in Figure 5b shows that MB adsorption at equilibrium is adequate for the Freundlich equation isotherm, since the correlation coefficient value (R^2^) is 0.93. The amount of MB dye adsorbed *q_e_* increased with increasing equilibrium concentrations *Ce*. The Freundlich parameters and the numerical fits of sorption data are shown in Table 5. It was observed that the values of 1/*n* obtained ranged at 0.392 that suggests the chemisorption’s process [49]. Comparing the two isotherms reveals that the correlation coefficients for the Langmuir isotherm is higher than the Freundlich isotherm and showed better fit onto prepared NAC. So, the dye adsorption process onto carboxylic functionalized activated carbon is believed to be a chemisorptive process rather than a physisorptive one [50]. Furthermore, the higher adsorption capacity of NAC indicates that the added carboxylic functional groups are most likely involved in the process of chemisorption. Besides, the acidic oxygenated groups may work as ion-exchange for heavy metal cations due to the anionic nature of oxygen groups [51]. 

### 3.4. Adsorption Model of NAC Material 

It was suggested that there are no strong changes present in the textural properties of the optimum prepared nano-activated carbon compared with El-Maghara coal. Moreover, there is no considerable relationship between the material textural properties and its behavior as an adsorbent material for dye removal. Accordingly, the textural properties of the prepared NAC is difficult to detect through its dye adsorption behavior [52]. Activated carbons are amphoteric materials; thus, their surfaces might be positively or negatively charged depending on the solution pH. At pH > pHpzc, the carbon surface gained negative charge favoring the adsorption of cationic species like MB dye. On the other hand, adsorption of anionic species will be favored at pH ˂ pHpzc [53]. The pHpzc is also an indicator for oxidation of carbons surface, since it points out an increase in surface acidity/basicity after treatment. In the present study, the pH pzc of the prepared NAC was recorded as 4.8, which is homogeneous with other values stated in the literature [54]. If the electrostatic interactions were the main adsorption mechanism, basic dyes would be predicted to have greater affinity onto the prepared nano-activated carbon (pHpzc ˂ pH). Nonetheless, two parallel adsorption mechanisms, the first included electrostatic interactions and the second is dispersive interactions that are testified to establish the organic dyes adsorption onto NAC [55].

The interaction of the surface of activated carbon and dye molecules is predicted to occur between the free electrons at aromatic rings of dye and multiplied bonds and the delocalized π electrons of the oxygen-free Lewis basic sites at NAC. Where, the broad peaks appeared around 3400–3800 cm^−1^ and 1463 cm^−1^ that are ascribed to O–H stretching vibration involved in hydrogen bonds at NAC spectra before dye adsorption disappeared after dye adsorption process, as indicated in Figure 6. This FTIR spectra of NAC after dye adsorption returned to the interaction between the negatively charged hydroxyl ions groups at NAC and the positively charged MB dye molecules. This result confirms that all the active adsorption sites onto carbon samples were consumed at the dye adsorption. As well, a new peak was illustrated at the carbon sample after dye adsorption at 1236 cm^−1^ that was assigned to C–N stretching vibration of the methylene blue dye [55]. These results confirm the MB dye adsorption onto the prepared nano-activated carbon after completing the adsorption process. 

As methylene blue (basic dye) dissolved in water, their molecules are positively charged; thus it was expected that it would be preferably adsorb onto acid modified NAC. The oxidation of AC by nitric acid has been found to impart oxygen groups onto the surface of produced nano-activated carbon in slightly different manners, and this has the ability to adapt the surface chemistry of a carbon material that represents one of the major reasons that viewed NAC as a promising adsorbent [52].

### 3.5. Assessment of Optimum Prepared NAC and Its Parent Raw Coal for Multi Heavy Metals Adsorption from Real Petrochemicals Discharged Wastewater

Most of the industrial processes generate effluents with more than one metal ion. So, the adsorption performance of both optimum prepared NAC and Raw Coal (RC) on the treatment of the petrochemical discharged wastewater that contaminated with six heavy metals was evaluated to determine the metal competition at the substrate binding sites. Table 6 and Figure 7 summarize the results of the batch treatment process using the two materials. It was observed that the removal efficiency of lead and zinc was much higher than that for copper, nickel, iron, and manganese that followed the order of Pb^2+^ (0.119 nm) > Zn^2+^ (0.074 nm) > Cu^2+^ (0.071 nm) > Ni^2+^ (0.070 nm) > Fe^2+^ (0.069 nm) > Mn^2+^ (0.081 nm) [53]. This metal ions adsorption affinity order was similar for both the Raw Coal (RC) and prepared NAC; however, as indicated from Table 6 and Figure 7, the NAC showed promising adsorption results compared with RC. This result may be due to the large surface area and porous structure of prepared NAC beside its carboxylic acid and quinone functional groups that were introduced onto its surface after nitric acid oxidation. However, the results of the heavy metal adsorption order sequence may be returned to the metal ions competition at the same adsorption sites. Additionally, the priority of metal ions adsorption may be owed to the ionic radii of the metal ions, where both lead and zinc ions characterized by their high ionic radii compared with other studied metal ions. So, it is expected that the largest ionic radii and molecular weight of metal ion may have the highest adsorption selectivity onto the studied carbon-based materials [54]. In spite of the high adsorption affinity of optimum prepared NAC for the different studied heavy metals, however, the prepared material attains higher adsorption affinity for MB dye compared with studied heavy metals. Where, it was indicated from Figure 7 that Fe metal represents the favorite adsorbed metal onto the prepared NAC with adsorption removal of 60% compared with 91.3% adsorption for MB that is recorded on Table 3 after the equilibrium time. These results may be owed to the presence of different metal ions associated with Fe ion during studying the heavy metal adsorption process onto NAC. These multi-metal ions are competing with each other to interact with the active sides of the NAC that decreases the adsorption of each metal ion. However, in the MB dye adsorption process, the presence of dye molecules only increases their change to be interacted with the active sites onto NAC that improves the dye adsorption affinity compared with the multi-metal adsorption. 

These results noted that there is a significant contribution of electrostatic interaction between the negative charges groups at NAC surface and the positive charges of heavy metals ions and/or MB as an example of cationic dye. Also, hydrogen bonding and chelation may occur between NAC and divalent heavy metal ions, although, their organic pollutant adsorption performances decrease with acid interaction [55]. This conclusion was supported by the FTIR result of adsorbed dye onto NAC presented in Figure 6. 

Consequently, the prepared NAC is suitable for the removal of heavy metal ions from wastewater discharged from petrochemical industries.

## 4. Conclusions

The adsorption profile of any process is highly reliant on the features of the adsorbent utilized; a surface modification procedure often being required to optimize the adsorbent properties. In this study, nano-activated carbon prepared from Maghara coal is sensitive toward nitric acid modification. The aim of this chemical treatment was to improve the hydrophilic nature of the carbon surface that increases its adsorption capacity toward wastewater contaminates. The results revealed that the nitric acid modification process of raw coal increases both the pore volume and surface area of produced nano-activated carbon. 

The optimum nitric acid modified activated carbon is produced at 10% nitric acid concentration at mixing ratio of HNO_3_/AC of 20 and 2 h oxidation time. The oxidation treatment increases the total acidity due to the conjugated surface oxide groups such as carboxyl and quinone that was detected from FTIR for the prepared material. The adsorption isotherms were evaluated using Langmuir and Freundlich models. The Langmuir model provided a better fit for dye adsorption onto NAC material, indicating that the adsorption occurs with the formation of a monolayer. This result confirmed that the cationic ions adsorption on the prepared activated carbons is improved by the existence of surface acid groups. The nitric acid modification optimizes the existing properties of prepared nano-activated carbon to be suitable for metal ion removal from polluted industrial wastewater.

## Figures and Tables

**Figure 1 materials-13-02498-f001:**
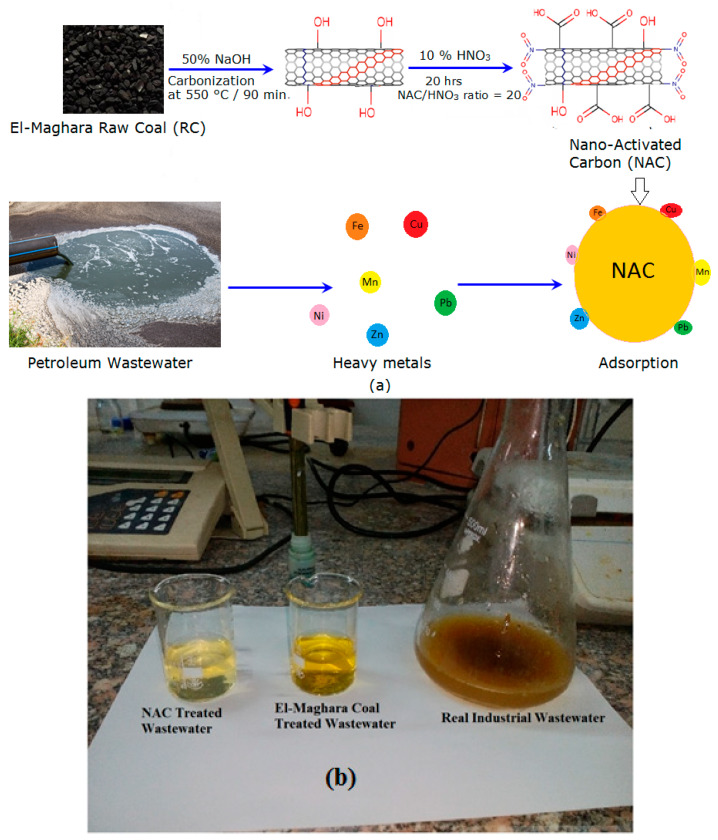
(**a**) Schematic representation of the chemical modification of El-Maghara coal for nano- activated carbon (NAC) fabrication, and (**b**) Digital image of the real industrial wastewater before and after treatment with carbon materials.

**Figure 2 materials-13-02498-f002:**
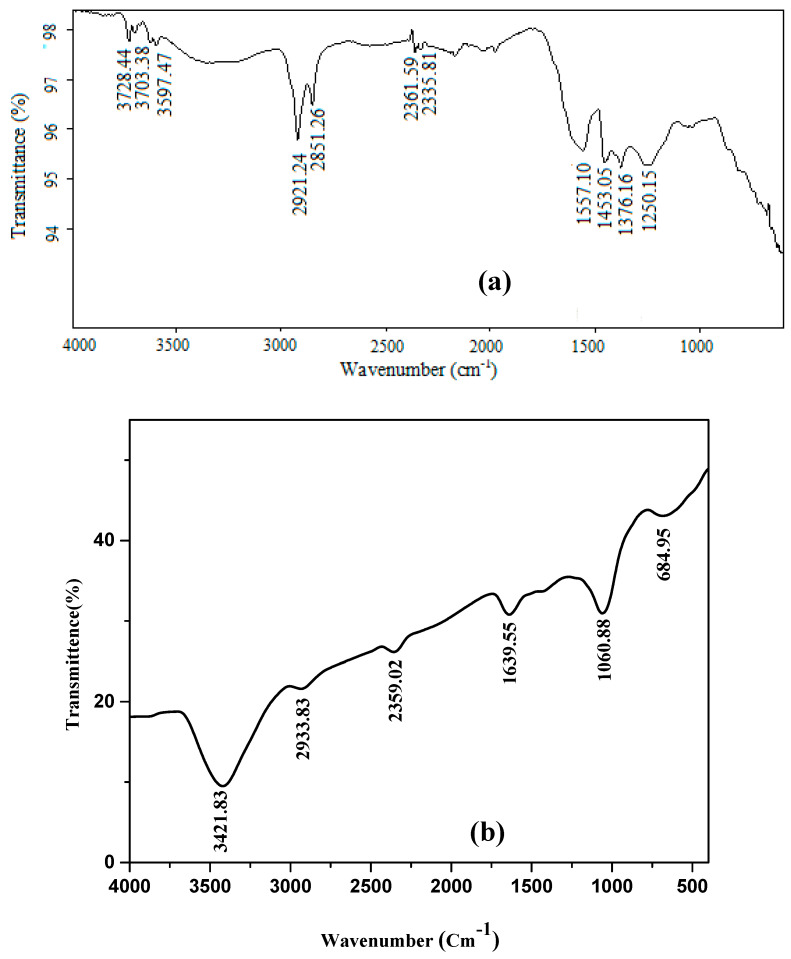
FTIR spectrum of (**a**) El-Maghara mine coal, and (**b**) nano-activated carbon.

**Figure 3 materials-13-02498-f003:**
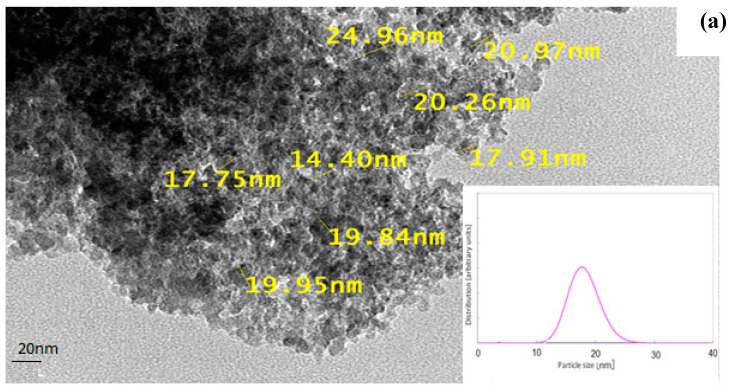
(**a**) TEM micrograph and particle size distribution, (**b**) SEM micrograph and (**c**) Raman spectrum of prepared nano-activated carbon (NAC).

**Figure 4 materials-13-02498-f004:**
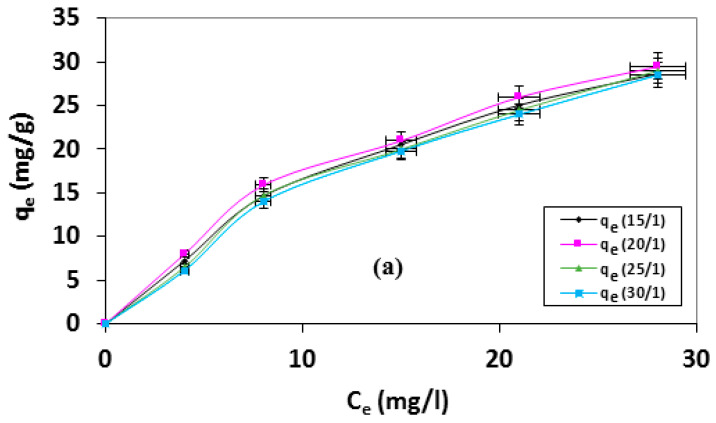
Adsorption isotherms of MB at 25 ± 2 °C onto nitric acid treated AC at different (**a**) nitric acid/AC ratios; (**b**) oxidation times; and (**c**) nitric acid concentrations (oxidation time = 2 h, nitric acid concentration = 30%, nitric acid/AC ratio = 20/1).

**Figure 5 materials-13-02498-f005:**
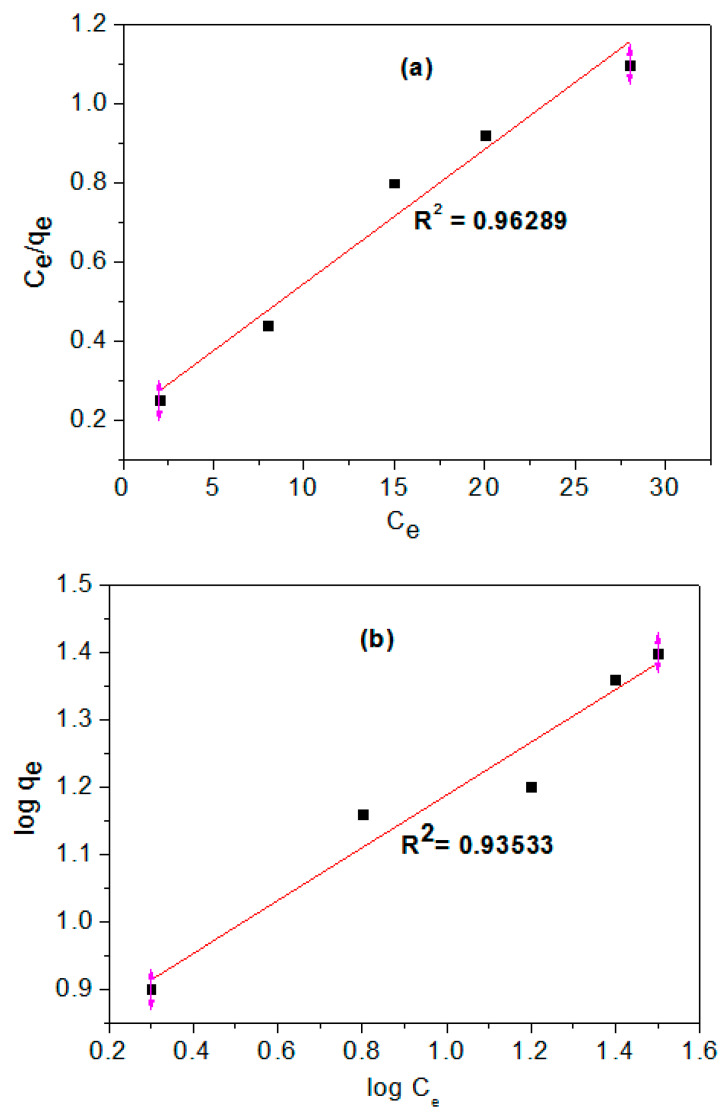
(**a**) Langmuir and (**b**) Freundlich equilibrium isotherms of MB sorption onto optimum prepared NAC.

**Figure 6 materials-13-02498-f006:**
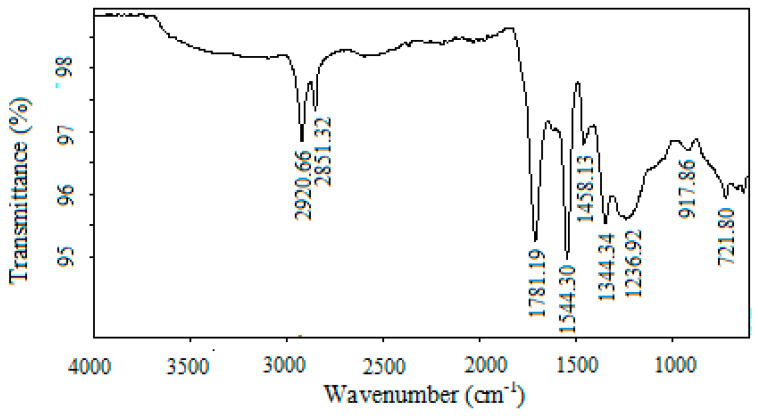
FTIR spectrum nano-activated carbon after dye adsorption.

**Figure 7 materials-13-02498-f007:**
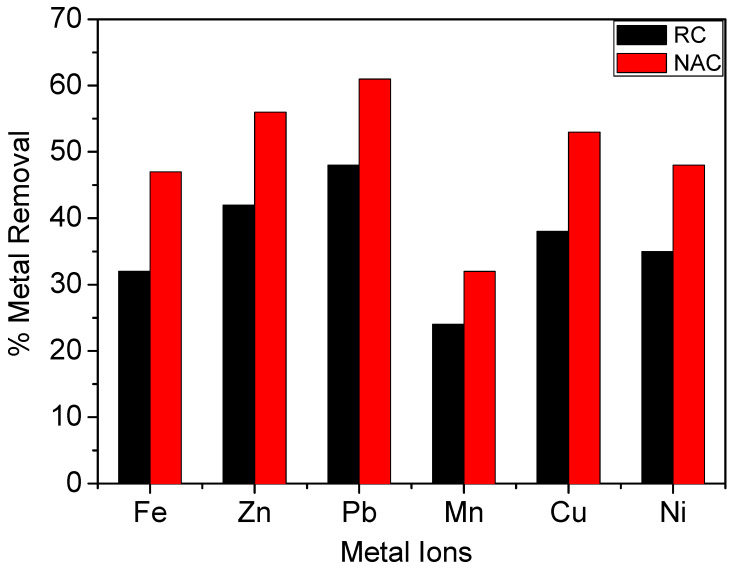
Schematic diagram for heavy metals decontamination from discharged petrochemical wastewater using raw coal and nano-activated carbon (0.2 g/L carbon sample, contact time 2 h, temperature 25 ± 2 °C).

**Table 1 materials-13-02498-t001:** Type of coal and percentage of carbon.

Coal Type (Rank)	% Carbon	H.V (BTU/Ib)
Anthracite	86–98	15,000
Bituminous	45–86	10,500–15,500
Sub bituminous	35–45	8300–13,000
Lignite	25–35	4000–8300

**Table 2 materials-13-02498-t002:** Chemical analysis of Maghara coal.

Maghara Coal Content	Value (%)
Carbon	71.44
Hydrogen	6.62
Oxygen	9.1
Nitrogen	2.27
Sulphur	3.8

**Table 3 materials-13-02498-t003:** Determination of equilibrium time for methylene blue (MB) dye adsorption onto activated carbon.

Adsorption Time (Contact Time)	Final Dye Concentration (ppm)
0	30
10	21.1
20	14.2
30	8.6
40	3.1
50	2.8
60	2.6
75	2.5
90	2.5
105	2.4

**Table 4 materials-13-02498-t004:** Chemical composition of prepared NAC using EDX analysis.

Elements	Weight (%)	Atomic (%)
C	80.5	85.4
O	12.6	10.1
N	6.4	4.4
K	0.34	0.08
S	0.16	0.02

**Table 5 materials-13-02498-t005:** Langmuir and Freundlich parameters of adsorption process at optimum conditions.

Isotherm	Parameters	Value
Langmuir	*q_m_* (mg/g)	29.50
	*K_a_* (L/mg)	0.165
	R^2^	0.963
Freundlich	*K_f_* (mg/g(L/g)^1/n^)	2.217
	n	2.551
	R^2^	0.935

**Table 6 materials-13-02498-t006:** Heavy metal ions decontamination from petrochemicals discharged waste water using prepared NAC and raw coal (0.2 g/L carbon sample, 2 h, 25 ± 2 °C).

Metal Ion	Initial Concentration (ppm)	Final Concentration (RC)	Final Concentration (NAC)
Fe(II)	3.2	2.176	1.696
Ni(II)	0.664	0.4316	0.34528
Cu(II)	1.18	0.7316	0.5546
Zn(II)	2.45	1.176	1.078
Mn(II)	1.75	1.33	1.19
Pb(II)	1.86	0.7254	0.8742

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
