# Peer review of "New Activated Carbon from Mine Coal for Adsorption of Dye in Simulated Water or Multiple Heavy Metals in Real Wastewater"

_materials, 2020, doi:10.3390/ma13112498_

Round 1

Reviewer 1 Report

The manuscript prepared a new kind of activated carbon (AC) from mine coal and treated the as-prepared AC using nitric acid. The adsorption performance towards methylene blue and heavy metal ions of the AC were investigated. The results are interesting and make contributions in environmental and chemical engineering. After carefully reviewing, I think this manuscript is suitable to publish on Materials. However, a major revision is recommended.

  1. There are a couple of grammar errors throughout the manuscript. The authors need to re-read the manuscript carefully and revise all grammar issues. A proof-reading by an English editor is recommended. The following lists some examples of such grammar errors:

Line 23, 24; Line 65, 66; Line 77, 78; Line 294, 295, etc.

  1. Nano-activated carbon is used for the AC prepared in the study. However, there is no solid evidence that the AC prepared is a nano-material. Although the TEM picture showed some nanoparticles structures, it is not enough to support that the AC is a nano-activated carbon.
  2. Line 47-49, polymer-enhanced ultrafiltration is also a promising technique to remove heavy metals and dyes. The following references may be helpful to improve the manuscript (Journal of Membrane Science 586 (2019) 53-83; Journal of Membrane Science 514 (2016) 229-240; Separation and Purification Technology 158 (2016) 124-136)
  3. In line 114, it stated that the equilibrium was obtained after 1 hour. However, there was no experiment showing that 1 hour was enough to reach equilibrium. It is recommended to add a preliminary test to investigate the equilibrium time.
  4. The format of the equations needs to be improved.
  5. Although the current structure of Results and Discussion looks fine, it is recommended to re-organize this section according to the following suggestions:
  • Sections 3.2, 3.3, and 3.4 could be merged into one section titled “Effect of oxidation treatment conditions on MB adsorption”
  • Section 3.7 could be put in the beginning of the discussion, and merged together with FTIR results.
  • Figure 5b is recommended to put into Experimental section.
  1. In Fig 1, all three FTIR spectrums could be put into one figure so that the difference between each sample would be clearer.
  2. Line 300, please explain what the pHpzc is. In addition, if the surface charge distribution of the AC is important for the adsorption, a Zeta potential test is recommended to further support the discussion here.
  3. In Fig 5a, there is no scale shown along y-axis.

Author Response

Reviewer 1:

The manuscript prepared a new kind of activated carbon (AC) from mine coal and treated the as-prepared AC using nitric acid. The adsorption performance towards methylene blue and heavy metal ions of the AC were investigated. The results are interesting and make contributions in environmental and chemical engineering. After carefully reviewing, I think this manuscript is suitable to publish on Materials. However, a major revision is recommended.

  1. There are a couple of grammar errors throughout the manuscript. The authors need to re-read the manuscript carefully and revise all grammar issues. A proof-reading by an English editor is recommended. The following lists some examples of such grammar errors:

Line 23, 24; Line 65, 66; Line 77, 78; Line 294, 295, etc.

Reply: all the manuscript was revised and improved.

  1. Nano-activated carbon is used for the AC prepared in the study. However, there is no solid evidence that the AC prepared is a nano-material. Although the TEM picture showed some nanoparticles structures, it is not enough to support that the AC is a nano-activated carbon.

Reply: In order to confirm the nano-structure of prepared nanomaterials, both SEM micrograph and particle size analysis were done and added at the manuscript to support the TEM result.

  1. Line 47-49, polymer-enhanced ultrafiltration is also a promising technique to remove heavy metals and dyes. The following references may be helpful to improve the manuscript (Journal of Membrane Science 586 (2019) 53-83; Journal of Membrane Science 514 (2016) 229-240; Separation and Purification Technology 158 (2016) 124-136).

Reply: Authors added the suggested helpful references.

  1. In line 114, it stated that the equilibrium was obtained after 1 hour. However, there was no experiment showing that 1 hour was enough to reach equilibrium. It is recommended to add a preliminary test to investigate the equilibrium time.

Reply: the preliminary results of estimation the equilibrium time of dye adsorption onto prepared activated carbon was investigated at Table (3)

  1. The format of the equations needs to be improved.

Reply: The quality and format of the equations were improved.

  1. Although the current structure of Results and Discussion looks fine, it is recommended to re-organize this section according to the following suggestions:
  • Sections 3.2, 3.3, and 3.4 could be merged into one section titled “Effect of oxidation treatment conditions on MB adsorption”
  • Section 3.7 could be put in the beginning of the discussion, and merged together with FTIR results.
  • Figure 5b is recommended to put into Experimental section.

Reply: Sections 3.2, 3.3, and 3.4 were merged into one section titled “Effect of oxidation treatment conditions on MB adsorption”.

Section 3.7 was put in the beginning of the discussion, and merged together with FTIR results.

Figure 5b was put into Experimental section.

  1. In Fig 1, all three FTIR spectrums could be put into one figure so that the difference between each sample would be clearer.

Reply: According to your advice at your previous comment (comment 6), the manuscript was rearranged to be started with investigation the material properties (material characterization) that followed by investigation the effect of oxidation parameters then description the dye adsorption model on nano-activated carbon. So, the FTIR characterization was divided into 2 parts, the first part was inserted at the first section that deals with the material characterization to identify the materials function groups, the second part was added at the description of dye adsorption model onto nano-activated to clarify the interaction between the material and dye molecules. So, according to this arrangement of the manuscript, it seems very difficult to add the three FTIR spectra together at one figure.

  1. Line 300, please explain what the pHpzc is. In addition, if the surface charge distribution of the AC is important for the adsorption, a Zeta potential test is recommended to further support the discussion here.

Reply: The explanation of pHpzc was added at the material and method section. Regarding to the importance of charge surface distribution, as mention at the manuscript that activated carbon material was characterized with amphoteric properties, so, the change at the surface charge of the material due to the change of media pH has significant effect on the interaction between the dye molecules and material that was explained in details at the “adsorption model” section. On the other hand, the equipment of zeta potential is not available at the authors’ institutes.

9.In Fig 5a, there is no scale shown along y-axis

Reply: The scale of y-axis was added and the figure was improved.

Reviewer 2 Report

The study is clear and adequately described. The approach is some what conventional, but the results will be interest to some readers which interested in environmental science.

Author Response

Reviewer 2:

The manuscript Materials-734288 reports method of preparation of new active carbon from El-Maghara coal and its application in removal of dye methylene blue and heavy metal (Fe2+, Ni2+, Mn2+, Pb2+, Cu2+ and Zn2+). The paper reviewed contains interesting results. Unfortunately interpretation should be improved. In my opinion this article is worth to be published in Materials after major revision. I have some suggestions/questions which authors may consider prior to publication of this work:

  1. The aim of the work should be corrected. In its current form, it does not apply to all investigation undertaken in the paper. The authors did not refer to research related to heavy metal recovery. This is the main problem with this paper. The authors did not explain why they compare such different groups of impurities, i.e. dyes and metal ions. This issue should be discussed both in the introduction and in results discussion by comparing the results obtained for both groups.

Reply: The aim of the work was changed to investigate the comparison behavior of dye adsorption and multi heavy metal adsorption all over the manuscript and all necessary modifications at the material & methods and results and discussion sections were included at the modified manuscript.  

  1. The basic characterization of Maghara coal obtained from El-Maghara field coal, as well as wastewater delivered from Sidi-Kerir Petrochemicals Company (i.e. pH, acid contents), should be added to the manuscript.

Reply: The basic characteristics of the coal was analyzed and added at the modified manuscript at Table 2. Also, the main characteristics of the wastewater were added at the manuscript.

  1. Authors wrote that “The nitric acid treated sample was washed with hot water until it reached a constant pH of 6.5.” The information about this process should be improved. How many times the washing procedure was repeated? How many water was used in this process? This procedure generated pollution, and such source should be taken into account in the overall balance of the proposed wastewater treatment process.

Reply: The details about the washing process for production activated carbon modified by nitric acid was added at the modified manuscript. However, we cannot consider the amount of washing water represents a pollution source, where each 0.5g sample produced was washed three times using centrifugation force  with total water amount of about 180ml which represents comparatively low amount to the total daily amount of discharged wastewater.

  1. Methylene blue normally is characterized by an absorption band at high energy (π-π* of benzene ring) and a band at low energy around 660-670 nm (moving according to the pH of the solution) and corresponding to n-π* transitions (n is the free doublet on the nitrogen atom of C=N bond and free doublet of S atom on S=C bond). The peak at 605 nm is not a band but a shoulder and it correspond to a vibrionic transition 0-1 (level 0 of ground state to level 1 of the excited state). Thus why did author use 650 nm to determine the dye concentration? Was it check, with the influence of acid concertation? The methylene blue spectra, with the all considered solution (i.e. nitric acid concentration), should be added to confirm the correctness of the assumptions.

Reply: Thank you for your comment, there is a typing mistake at the manuscript during writing the maximum peak for methylene blue adsorption. All work and detections for Methylene blue concentrations were takes place at 665nm not 650nm and the standard curve used for determination the remaining dye concentration after adsorption process was takes place at  665nm. So, this writing error was corrected.

  1. The batch experiments with dye are differ to heavy metal ions, in the first case the time is one hour, and in second one two hours. This difference, and the should be described in more detail.

Reply: The difference between the contact time for heavy metals and dye removal was due to for the removal process of dye there is only one type of pollutant that has high change to be adsorbed onto the active sides of modified activated carbon. However, the case for the multi-metal removal is different, where, there is a high completion between each metal ion to be adsorbed onto the active sites at the adsorbent material. This behavior was proofed experimentally, where, the adsorption performance for the prepared material was tested against both dye removal and multi-metal removal at contact time of 1 hour. However, at this treatment conditions the recorded metal removal was very poor, so, the operating contact time was selected to be 2 hours for metal ions removal to be able to differentiate between the selectivity of the material towards the different studied metals.

The reason for selection the 2 hours as contact time for test the adsorption performance of the modified activated carbon material toward the decontamination of the muti-metal ions polluted wastewater was added at the modified manuscript.

  1. The abbreviations in equations 2 and 3 should be explain, and the units should be added in the manuscript.

Reply: The equations abbreviations were inserted at the modified manuscript.

  1. The section 3.1 should be rewritten. At first the quality of Figures 1 a-c should be improved, and why Figure b is different from Figures A and C. Moreover the interpretation requires the correction. For example authors wrote that: i. “The broad peaks around 3400–3800 cm-1 and 1463 cm-1at the two carbon samples spectra (…)”, but there are no peak at 1463 cm-1, as well as broad peaks at two samples; ii. “The peaks presented at the two carbon materials at 917, 185 721,661 and 627 cm-1 are assigned to the aromatic ring's presence” – in which two materials it is possible to see these peaks?; iii. “However, a new peak was illustrated at the carbon sample after dye adsorption at 1236 cm-1 that assign to C-N stretching vibration at methylene blue dye” – it’s true that there is a peak at 1236 in Figure 1C, but in Figure 1 A there is a peak at 1250 cm-1. It is the same or not? If not what is at 1250 cm-1.

Reply: This section was re-written completely and the figures quality was improved and the necessary interpretations and corrections for peaks definitions and their appearance place were corrected at the modified manuscript. Regarding to the difference between Figure b and A&C, this difference is expected because Figure A includes a combination between activated carbon + some organic compounds such as hemi-cellulosic compounds and lignin materials that were removed after formation nano-activated carbon (Figure b). Also, Figure C includes a combination between activated carbon + organic compound which is organic dye. So, it was expected that the pure activated carbon that is free completely from the organic compounds which is represented by Figure b will be differ from both other 2 figures that include combinations between activated carbon and organic molecules (Figures a &c).  

  1. Due to these objections presented in point no 4 it is uncertain whether the calculations made in the chapters 3.2 – 3.4 are correct. Moreover the results on the Figures 2 A-C are very similar. Without showing the error bars it is not possible to assess whether the considerations presented in the paper, are correct, whether the effect is important or if the differences are only within the error limits.

Reply: As previously declared at point 4, the mistake wasn’t at the work, however, it is only typing mistake at the number, also, as previously declared that all work and measurements were takes place at 665nm and each experiment and measurements were repeated 3 times and the average measure was assigned. Moreover, the similarity between the figures is expected because we present the effect of synthetic parameter against the change at the material adsorption capacity, so, it was expected a small division at the material capacity with the change in the modification synthesis process, where we utilized just one material and it is expected that the change in variation in its adsorption capacity with variation modification conditions will not be very large.  In spite of there is a small change in the material adsorption capacity with change at the synthesis and modification process, however, it is still detectable and varied. Moreover, according to your advice to declare this small variation at material adsorption capacity, the error bars were added at the modified manuscript. 

  1. Authors use subscript for Ce, qe, but unfortunately in Figures not any more. This should be standardized in the manuscript

Reply: the subscript for Ce, qe, was added at the figures

  1. The metal removal is very low, moreover difference in adsorption ability for NAC and RC is also small (the scale should be added to the Figure 5 A!). Authors should emphasize the added value of their solution

Reply: the meal removal is very low as discussed previously at point 5 that the presence of multi-metal ions during the adsorption process hinder each other and decrease the rate of metal adsorption for that 2hours contact time was selected to study the adsorption process compared with one hour for dye adsorption. Regarding to the scale of Figure 5 A, the figure was re-draw to present the difference in metal removal for both NAC & RC

Reviewer 3 Report

Review of paper “New activated carbon from mine coal for adsorption of dye in simulated water or multiple heavy metals in real wastewater” prepared by Marwa Elkady, Hassan Shokry, and Hesham Hamad.

The manuscript Materials-734288 reports method of preparation of new active carbon from El-Maghara coal and its application in removal of dye methylene blue and heavy metal (Fe2+, Ni2+, Mn2+, Pb2+, Cu2+ and Zn2+). The paper reviewed contains interesting results. Unfortunately interpretation should be improved. In my opinion this article is worth to be published in Materials after major revision. I have some suggestions/questions which authors may consider prior to publication of this work:

  1. The aim of the work should be corrected. In its current form, it does not apply to all investigation undertaken in the paper. The authors did not refer to research related to heavy metal recovery. This is the main problem with this paper. The authors did not explain why they compare such different groups of impurities, i.e. dyes and metal ions. This issue should be discussed both in the introduction and in results discussion by comparing the results obtained for both groups.
  2. The basic characterization of Maghara coal obtained from El-Maghara field coal, as well as wastewater delivered from Sidi-Kerir Petrochemicals Company (i.e. pH, acid contents), should be added to the manuscript.
  3. Authors wrote that “The nitric acid treated sample was washed with hot water until it reached a constant pH of 6.5.” The information about this process should be improved. How many times the washing procedure was repeated? How many water was used in this process? This procedure generated pollution, and such source should be taken into account in the overall balance of the proposed wastewater treatment process.
  4. Methylene blue normally is characterized by an absorption band at high energy (π-π* of benzene ring) and a band at low energy around 660-670 nm (moving according to the pH of the solution) and corresponding to n-π* transitions (n is the free doublet on the nitrogen atom of C=N bond and free doublet of S atom on S=C bond). The peak at 605 nm is not a band but a shoulder and it correspond to a vibrionic transition 0-1 (level 0 of ground state to level 1 of the excited state). Thus why did author use 650 nm to determine the dye concentration? Was it check, with the influence of acid concertation? The methylene blue spectra, with the all considered solution (i.e. nitric acid concentration), should be added to confirm the correctness of the assumptions.
  5. The batch experiments with dye are differ to heavy metal ions, in the first case the time is one hour, and in second one two hours. This difference, and the should be described in more detail.
  6. The abbreviations in equations 2 and 3 should be explain, and the units should be added in the manuscript.
  7. The section 3.1 should be rewritten. At first the quality of Figures 1 a-c should be improved, and why Figure b is different from Figures A and C. Moreover the interpretation requires the correction. For example authors wrote that: i. “The broad peaks around 3400–3800 cm−1 and 1463 cm-1 at the two carbon samples spectra (…)”, but there are no peak at 1463 cm-1, as well as broad peaks at two samples; ii. “The peaks presented at the two carbon materials at 917, 185 721,661 and 627 cm-1 are assigned to the aromatic ring's presence” – in which two materials it is possible to see these peaks?; iii. “However, a new peak was illustrated at the carbon sample after dye adsorption at 1236 cm−1 that assign to C-N stretching vibration at methylene blue dye” – it’s true that there is a peak at 1236 in Figure 1C, but in Figure 1 A there is a peak at 1250 cm-1. It is the same or not? If not what is at 1250 cm-1.
  8. Due to these objections presented in point no 4 it is uncertain whether the calculations made in the chapters 3.2 – 3.4 are correct. Moreover the results on the Figures 2 A-C are very similar. Without showing the error bars it is not possible to assess whether the considerations presented in the paper, are correct, whether the effect is important or if the differences are only within the error limits.
  9. Authors use subscript for Ce, qe, but unfortunately in Figures not any more. This should be standardized in the manuscript.
  10. The metal removal is very low, moreover difference in adsorption ability for NAC and RC is also small (the scale should be added to the Figure 5 A!). Authors should emphasize the added value of their solution.

Author Response

The manuscript Materials-734288 reports method of preparation of new active carbon from El-Maghara coal and its application in removal of dye methylene blue and heavy metal (Fe2+, Ni2+, Mn2+, Pb2+, Cu2+ and Zn2+). The paper reviewed contains interesting results. Unfortunately interpretation should be improved. In my opinion this article is worth to be published in Materials after major revision. I have some suggestions/questions which authors may consider prior to publication of this work:

  1. The aim of the work should be corrected. In its current form, it does not apply to all investigation undertaken in the paper. The authors did not refer to research related to heavy metal recovery. This is the main problem with this paper. The authors did not explain why they compare such different groups of impurities, i.e. dyes and metal ions. This issue should be discussed both in the introduction and in results discussion by comparing the results obtained for both groups.

Reply: The aim of the work was changed to investigate the comparison behavior of dye adsorption and multi heavy metal adsorption all over the manuscript and all necessary modifications at the material & methods and results and discussion sections were included at the modified manuscript.  

  1. The basic characterization of Maghara coal obtained from El-Maghara field coal, as well as wastewater delivered from Sidi-Kerir Petrochemicals Company (i.e. pH, acid contents), should be added to the manuscript.

Reply: The basic characteristics of the coal was analyzed and added at the modified manuscript at Table 2. Also, the main characteristics of the wastewater were added at the manuscript.

  1. Authors wrote that “The nitric acid treated sample was washed with hot water until it reached a constant pH of 6.5.” The information about this process should be improved. How many times the washing procedure was repeated? How many water was used in this process? This procedure generated pollution, and such source should be taken into account in the overall balance of the proposed wastewater treatment process.

Reply: The details about the washing process for production activated carbon modified by nitric acid was added at the modified manuscript. However, we cannot consider the amount of washing water represents a pollution source, where each 0.5g sample produced was washed three times using centrifugation force  with total water amount of about 180ml which represents comparatively low amount to the total daily amount of discharged wastewater.

  1. Methylene blue normally is characterized by an absorption band at high energy (π-π* of benzene ring) and a band at low energy around 660-670 nm (moving according to the pH of the solution) and corresponding to n-π* transitions (n is the free doublet on the nitrogen atom of C=N bond and free doublet of S atom on S=C bond). The peak at 605 nm is not a band but a shoulder and it correspond to a vibrionic transition 0-1 (level 0 of ground state to level 1 of the excited state). Thus why did author use 650 nm to determine the dye concentration? Was it check, with the influence of acid concertation? The methylene blue spectra, with the all considered solution (i.e. nitric acid concentration), should be added to confirm the correctness of the assumptions.

Reply: Thank you for your comment, there is a typing mistake at the manuscript during writing the maximum peak for methylene blue adsorption. All work and detections for Methylene blue concentrations were takes place at 665nm not 650nm and the standard curve used for determination the remaining dye concentration after adsorption process was takes place at  665nm. So, this writing error was corrected.

  1. The batch experiments with dye are differ to heavy metal ions, in the first case the time is one hour, and in second one two hours. This difference, and the should be described in more detail.

Reply: The difference between the contact time for heavy metals and dye removal was due to for the removal process of dye there is only one type of pollutant that has high change to be adsorbed onto the active sides of modified activated carbon. However, the case for the multi-metal removal is different, where, there is a high completion between each metal ion to be adsorbed onto the active sites at the adsorbent material. This behavior was proofed experimentally, where, the adsorption performance for the prepared material was tested against both dye removal and multi-metal removal at contact time of 1 hour. However, at this treatment conditions the recorded metal removal was very poor, so, the operating contact time was selected to be 2 hours for metal ions removal to be able to differentiate between the selectivity of the material towards the different studied metals.

The reason for selection the 2 hours as contact time for test the adsorption performance of the modified activated carbon material toward the decontamination of the muti-metal ions polluted wastewater was added at the modified manuscript.

  1. The abbreviations in equations 2 and 3 should be explain, and the units should be added in the manuscript.

Reply: The equations abbreviations were inserted at the modified manuscript.

  1. The section 3.1 should be rewritten. At first the quality of Figures 1 a-c should be improved, and why Figure b is different from Figures A and C. Moreover the interpretation requires the correction. For example authors wrote that: i. “The broad peaks around 3400–3800 cm-1 and 1463 cm-1at the two carbon samples spectra (…)”, but there are no peak at 1463 cm-1, as well as broad peaks at two samples; ii. “The peaks presented at the two carbon materials at 917, 185 721,661 and 627 cm-1 are assigned to the aromatic ring's presence” – in which two materials it is possible to see these peaks?; iii. “However, a new peak was illustrated at the carbon sample after dye adsorption at 1236 cm-1 that assign to C-N stretching vibration at methylene blue dye” – it’s true that there is a peak at 1236 in Figure 1C, but in Figure 1 A there is a peak at 1250 cm-1. It is the same or not? If not what is at 1250 cm-1.

Reply: This section was re-written completely and the figures quality was improved and the necessary interpretations and corrections for peaks definitions and their appearance place were corrected at the modified manuscript. Regarding to the difference between Figure b and A&C, this difference is expected because Figure A includes a combination between activated carbon + some organic compounds such as hemi-cellulosic compounds and lignin materials that were removed after formation nano-activated carbon (Figure b). Also, Figure C includes a combination between activated carbon + organic compound which is organic dye. So, it was expected that the pure activated carbon that is free completely from the organic compounds which is represented by Figure b will be differ from both other 2 figures that include combinations between activated carbon and organic molecules (Figures a &c).  

  1. Due to these objections presented in point no 4 it is uncertain whether the calculations made in the chapters 3.2 – 3.4 are correct. Moreover the results on the Figures 2 A-C are very similar. Without showing the error bars it is not possible to assess whether the considerations presented in the paper, are correct, whether the effect is important or if the differences are only within the error limits.

Reply: As previously declared at point 4, the mistake wasn’t at the work, however, it is only typing mistake at the number, also, as previously declared that all work and measurements were takes place at 665nm and each experiment and measurements were repeated 3 times and the average measure was assigned. Moreover, the similarity between the figures is expected because we present the effect of synthetic parameter against the change at the material adsorption capacity, so, it was expected a small division at the material capacity with the change in the modification synthesis process, where we utilized just one material and it is expected that the change in variation in its adsorption capacity with variation modification conditions will not be very large.  In spite of there is a small change in the material adsorption capacity with change at the synthesis and modification process, however, it is still detectable and varied. Moreover, according to your advice to declare this small variation at material adsorption capacity, the error bars were added at the modified manuscript. 

  1. Authors use subscript for Ce, qe, but unfortunately in Figures not any more. This should be standardized in the manuscript

Reply: the subscript for Ce, qe, was added at the figures

  1. The metal removal is very low, moreover difference in adsorption ability for NAC and RC is also small (the scale should be added to the Figure 5 A!). Authors should emphasize the added value of their solution

Reply: the meal removal is very low as discussed previously at point 5 that the presence of multi-metal ions during the adsorption process hinder each other and decrease the rate of metal adsorption for that 2hours contact time was selected to study the adsorption process compared with one hour for dye adsorption. Regarding to the scale of Figure 5 A, the figure was re-draw to present the difference in metal removal for both NAC & RC

Finally, authors want to thank reviewers for their valuable comments which are improved the scientific value of the manuscript

Round 2

Reviewer 1 Report

Thanks for the reply. The present manuscript looks good.

Reviewer 3 Report

Review of manuscript Materials_2019_734288, after corrections

The manuscript “New activated carbon from mine coal for adsorption of  dye in simulated water or multiple heavy metals in real wastewater. ” has been corrected by authors: Marwa Elkady, Hassan Shokry and Hesham Hamad. All my suggestions have been taken into account. I have no further comments. In my opinion this version of manuscript is worth to be published in Materials.